# Quality of reporting of cranial irradiation techniques in randomized controlled trials of primary brain tumors: A systematic review

Teng Hwee Tan[1,2,3,4], Desiree Chen[1,2,3,4], Yu Yang Soon[1,2,3,4]*, Jeremy Chee Seong Tey[1,2,3,4]

1 Department of Radiation Oncology, National University Cancer Institute, Singapore, Singapore, 2 National University Hospital, Singapore, Singapore, 3 National University Health System, Singapore, Singapore, 4 National University of Singapore, Singapore, Singapore

* yu_yang_soon@nuhs.edu.sg

**Data Availability Statement:** The data are all contained within the paper and supporting information files.

## Abstract

### Background

To assess the quality of reporting of cranial irradiation (CR) techniques in randomized controlled trials (RCTs) of primary brain tumors.

### Methods

We searched PubMed and EMBASE for RCTs of primary brain tumors, published from January 1999 to November 2019 which included CR as one of the intervention arms. We assessed the initial RCTs report on whether they reported the prespecified ten criteria for CR technique adequately. Multivariable logistic regression was performed to determine the factors that were predictive of adequate quality of reporting.

### Results

We found 85 eligible trial reports. There was significant variability in the quality of reporting among the included studies. Total radiotherapy (RT) dose and fractionation schedule were reported adequately in more than 90% of the included trials. The organs at risk dose constraints, treatment verification procedures and presence or absence of deviations in RT treatment planning and delivery were reported adequately in less than 30% of included trials. Twenty-three trials (27%) reported seven criteria or more adequately. Multivariable analysis showed that trials conducted by cooperative groups, published RT quality assurance results and having a low risk of bias in the methodological quality have higher odds of having adequate quality in reporting of CR technique (judged as adequate reporting in seven criteria or more).

### Conclusions

The quality of reporting on CR techniques in the RCTs of primary brain tumors is variable and suboptimal. Guidelines should be introduced to improve clarity and ensure consistency in the quality of reporting.

**Funding:** The author(s) received no specific funding for this work.

**Competing interests:** The authors have declared that no competing interests exist.

## Introduction

Primary brain tumors are a heterogenous group of neoplasms arising from different parts of the CNS. The worldwide incidence rate of primary brain tumors is estimated to be 10.8 cases per 100,000 person-years [1]. In the United States, the incidence rate of primary brain tumors is estimated to be 28.6 cases per 100,000 adults and 5.6 cases per 100,000 children [2]. Malignant brain tumours form 33 percent of all primary brain tumours in adults and 65 percent in children [2].

Radiation therapy (RT) is one of the main treatment modalities for primary brain tumors. Adherence to treatment protocols in RT treatments is essential. A meta-analysis of eight randomized trials showed that the frequency of deviations from RT treatment protocol ranged from 8% to 71% and deviations from RT treatment protocol was associated with increased risk of treatment failure and death. This highlights that there should be a minimal deviation from the reported RT treatment protocol in clinical trials for RT treatment for patients in the real world [3]. One possible method to reduce the risk of deviation is to have accurate and clear reporting of the RT treatment in the manuscripts of these randomized trials. This will allow the radiation treatment team in the community to accurately reproduce the RT treatment utilized in these randomized trials.

Several studies have suggested that the quality of reporting of RT treatment were inconsistent across randomized trials of lymphoma, head and neck, prostate and lung cancers [4–7]. However, the evidence on the quality of reporting of cranial irradiation (CR) technique in randomized trials of primary brain tumors and factors that may predict the quality of reporting is unknown. It is helpful for the readers to know what are the characteristics of a trial report that are predictive for adequate quality in reporting of cranial RT technique. This will help readers to be more confident in the results reported by the trial investigators if there is a clear understanding of how radiation therapy treatment is delivered.

Thus, this study aims to determine the quality of reporting of the CR technique in the randomized trials of primary brain tumors and the factors that may predict the quality of reporting.

## Methods

### Trial eligibility criteria

This study included the full publication of randomized trials of pediatric or adult patients with histologically or radiologically proven malignant primary brain tumors. Trials that include brain metastases were excluded as they were not primary brain tumors. Trials that include primary central nervous system lymphoma were specifically excluded as the cranial irradiation technique that is used in the treatment of primary central nervous system lymphoma is the standard whole-brain radiation therapy technique that is well described in standard radiation oncology textbook and it is unlikely that the trial investigators will describe the standard whole-brain radiation therapy technique in detail in their trial reports. We used the initial trial report for assessing the quality of CR technique reporting. We used the trial protocol in the assessment of the quality of CR technique reporting if they were referenced in the trial report or provided as supplementary materials with the trial report.

### Search strategy

We identified the trials by searching PubMed and EMBASE from January 1999 to November 2019. This time period was chosen as we would like to know if there were any significant changes in the quality of CR technique reporting over twenty years. The search strategy included the medical subject headings (MESH) terms and its synonyms for "brain neoplasms"

and "radiotherapy", limited to randomized trials (S1 Table). The synonyms were searched as key words in the titles and abstracts. The results were then hand searched for eligible trials. Also, the reference lists of selected trials were screened for any other relevant trials.

## Selection of trials and data extraction

Three reviewers independently assessed the abstracts' eligibility identified by the search. The full-text article of any trial that appeared to meet the inclusion criteria was retrieved for closer examination. Any disagreements were resolved through a discussion.

The same reviewers use a standardized data collection form to extract the data independently. Data retrieved from reports include publication details, risk of bias in the methodologic quality assessment, and trial characteristics such as type of study population (pediatric vs adults), type of primary brain tumor histology (glioma vs others), cooperative group trial (yes vs no), sample size and types of the primary outcome (overall survival vs others).

## Methodological quality assessment

We assessed the methodologic quality using the RoB2 tool which assesses the risk of bias in five domains namely: randomization process, deviations from the intended interventions, missing outcome data, measurement of the outcome and selection of the reported result [8]. The overall risk of bias was determined based on the reviewers' judgement for each of the domains. An overall "low risk of bias "score is given when the study is judged to be at low risk of bias for all domains. An overall "some concerns" score is given when the study is judged to raise some concerns in at least one domain, but not to be at high risk for any domain. An overall "high risk of bias" score is given when the study is judged to be at high risk of bias in at least one domain.

## Quality assessment of reporting of cranial irradiation treatment

We assessed the quality of CR reporting according to the prespecified ten quality measure criteria (Table 1): radiotherapy dose prescription method, radiotherapy dose-planning procedures, the organ at risk dose constraints, target volume definition, immobilization procedures,

**Table 1. Adequacy definition of radiotherapy reporting criterion.**

| Criterion | Adequacy definition |
|---|---|
| Radiotherapy dose prescription method | For 3-dimensional conformal technique–the prescription point must be described |
| | For intensity modulated or arc therapy–the volume based dose prescription must be described. |
| Radiotherapy dose-planning procedures | Describe either as forward or inverse planning |
| Organ at risk dose constraints | Describe at least one organ at risk dose constraints |
| Target volume definition | At least the clinical target volume must be described |
| Immobilization procedures | Describe immobilization procedures such as use of stereotactic frame |
| Treatment verification procedures | Describe at least one treatment verification procedure such as portal imaging, or cone beam CT |
| Total radiation dose | Describe the total dose and dose per fraction |
| Fractionation schedule | Describe the number of fractions per day, fractions per week and total number of fractions |
| Conduct of quality assurance | Report whether quality assurance was conducted |
| Deviation in the radiation treatment planning and delivery | Report if there is any deviations from the radiation treatment planning and delivery |

treatment verification procedures, total radiation dose, fractionation schedule, the conduct of quality assurance, deviation in radiation treatment planning and delivery. These criteria were selected as they were highlighted in previous publications as important parameters that need to be reported clearly to ensure that radiation therapy treatment can be reproduced accurately [6, 7]. A priori, we defined a trial as having adequate quality reporting if seven or more criteria were reported adequately.

### Predictors for adequate quality in the reporting of CR technique

We have prespecified the following variables for investigation as potential predictors for adequate quality in the reporting of CR technique. They include type of study population (pediatric versus (vs) adult), year of publication, cooperative group trial (yes vs no), region where trial was conducted (North America vs others), primary outcome (overall survival vs others), sponsorship of trial (none or not reported vs non-industry vs industry), sample size, published in radiotherapy focused journal (yes vs no), trial protocol published (yes vs no), quality assurance results published (yes vs no), trial question (radiotherapy focused vs non radiotherapy focused), listed in trial registry (yes vs no or not reported), impact factor of journal for the year of publication, type of primary brain tumors (high grade glioma vs others), trial met its pre-specified endpoint (yes vs no) and risk of bias in methodological quality (some concerns vs low risk).

### Statistical analysis

The descriptive statistics were presented as percentages. We used a backward stepwise multivariable logistic regression model approach to identify the predictors and its associated odds ratio for adequate quality in reporting of CR technique. This approach allows all the possible explanatory predictors to be first entered in the model and at each step, we allowed the explanatory predictors to be gradually eliminated from the regression model when its p value is more than 0.2 to find the most parsimonious model that best explains the data. Predictors with p value less than 0.05 in the multivariable logistic regression were considered statistically significant. We prefer this backward stepwise approach because it reduces the number of predictors and may help to reduce multi-collinearity problem and resolve overfitting. All potential predictors except for year of publication, sample size and impact factor of the journal for the year of publication were analysed as categorical variables. All statistical analysis was performed using STATA (version16.0, StataCorp).

### Protocol and registration

This study does not have a published protocol and is not registered in any registry.

## Results

### Selection of trials

We identified 85 eligible trials as summarized in Fig 1. We screened the titles and abstracts of 512 articles and excluded 383 articles as they did not meet the inclusion criteria. We retrieved 129 full text articles for assessment of eligibility. We further excluded 40 articles as they did not have the population or interventions of interest. We also excluded additional four articles as they were different reports of the same trial.

### Characteristics of trials

The characteristics of the 85 included trials were summarized in Table 2. Majority of trials (more than 80%) focused on an adult population and high grade glioma. Two-thirds of the

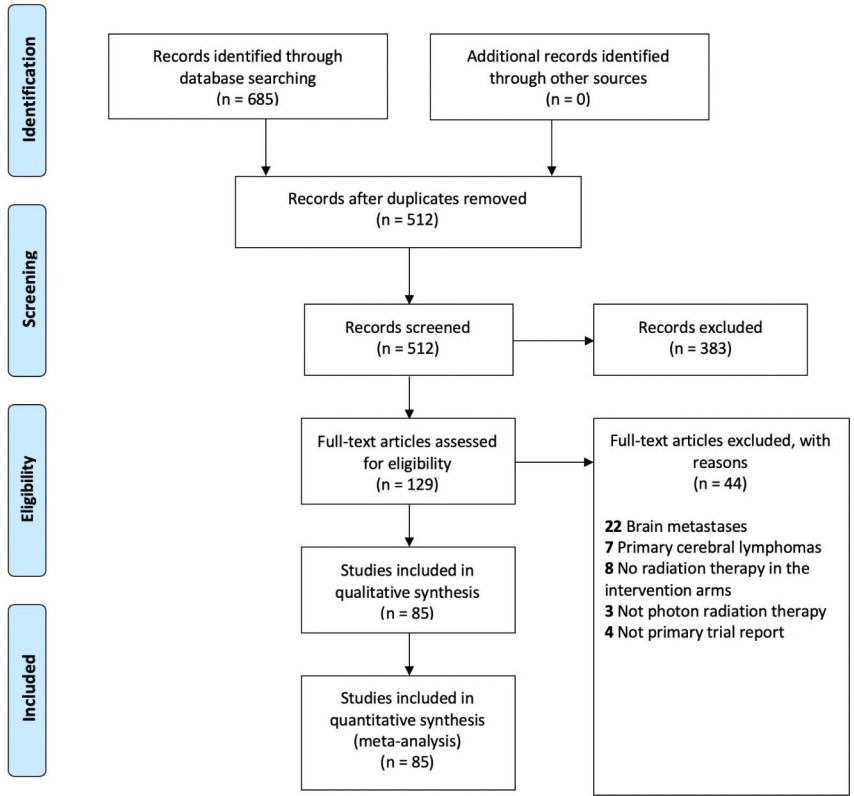

**Fig 1. Results of search strategy.**

trials used overall survival as primary endpoint. Trial protocol and QA results were only published in less than 20% of the trials.

## Quality of cranial irradiation reporting

There was significant variation in the quality of CR reporting among the included trials (Table 3 and Fig 2). Less than a third of the included trials reported the organ at risk dose constraints, immobilization procedures, treatment verification procedures and deviation in the radiation treatment planning and delivery criteria adequately. More than 75% of the included trials reported target volume definition, total radiation dose and fractionation schedule criteria adequately. Twenty seven percent (23/ 85) of trials reported seven criteria or more adequately i.e. these trials were judged to have adequate quality in reporting of CR treatment.

## Factors associated with adequate quality reporting

Multivariable logistic regression showed that trials conducted by cooperative groups, published QA results and have low risk of bias in its methodological quality were more likely to have adequate quality in the reporting of CR technique (Table 4). The odds of having adequate quality in reporting of CR technique among trials conducted by cooperative group were 4.65 times that of non-cooperative group trials (odds ratio (OR) 4.65, 95% confidence interval (CI) 1.13 to 19.11, P value (P) = 0.033). The odds of having adequate quality in reporting of CR technique for trials that published their QA results were 8.5 times that of trials that did not publish their QA results (OR 8.50, 95% CI 1.87 to 38.56, P = 0.006). The odds of having adequate quality in reporting of CR technique among trials with low risk of bias in the

**Table 2. Characteristics of included studies.**

| Characteristics | Trials (N = 85) | |
| --- | --- | --- |
| | N | % |
| Study population | | |
| Pediatric | 13 | 15 |
| Adult | 72 | 85 |
| Year of publication | | |
| 1999–2008 | 39 | 46 |
| 2009–2019 | 46 | 54 |
| Cooperative group | | |
| No | 35 | 41 |
| Yes | 50 | 59 |
| Region | | |
| North America | 29 | 34 |
| Others | 56 | 66 |
| Primary outcome | | |
| Overall survival | 58 | 68 |
| Others | 27 | 32 |
| Sponsorship | | |
| No or not reported | 18 | 21 |
| Non-industry | 49 | 58 |
| Industry | 18 | 21 |
| Sample size | | |
| ≤200 | 54 | 64 |
| >200 | 31 | 36 |
| Published in radiotherapy focused journals | | |
| Yes | 10 | 12 |
| No | 75 | 88 |
| Trial protocol published | | |
| Yes | 17 | 20 |
| No | 68 | 80 |
| QA results published | | |
| Yes | 14 | 16 |
| No | 71 | 84 |
| Trial question | | |
| Radiotherapy focused | 24 | 28 |
| Non-radiotherapy focused | 61 | 72 |
| Listed in trial registry | | |
| Yes | 32 | 38 |
| No or not reported | 53 | 62 |
| Impact factor of journal for the year of publication | | |
| ≤15 | 63 | 74 |
| >15 | 22 | 26 |
| Histology of primary brain tumors | | |
| High grade gliomas | 71 | 83 |
| Others | 14 | 17 |
| Risk of bias in methodologic quality | | |
| Low risk | 61 | 72 |
| Some concerns | 24 | 28 |
| Met prespecified primary endpoint | | |
| Yes | 59 | 69 |
| No | 26 | 31 |

**Table 3. Quality of cranial radiotherapy technique reporting (number of trials that reported each criterion adequately).**

| Criterion | No. of trials which reported this criterion adequately | % of trials, which reported this criterion adequately |
|---|---|---|
| Radiotherapy dose prescription method | 29 | 34 |
| Radiotherapy dose-planning procedures | 30 | 35 |
| Organ at risk dose constraints | 20 | 24 |
| Target volume definition | 64 | 75 |
| Immobilization procedures | 27 | 32 |
| Treatment verification procedures | 24 | 28 |
| Total radiation dose | 83 | 98 |
| Fractionation schedule | 79 | 93 |
| Conduct of quality assurance | 30 | 35 |
| Deviation in the radiation treatment planning and delivery | 15 | 18 |

methodological quality is 10 times that of trials with some concerns in the methodological quality (OR 10, 95% CI 1.23 to 100, P = 0.031).

## Discussion

This study demonstrated that there is significant variation in the quality of reporting cranial irradiation technique in randomized trials of primary brain tumors published over a twenty-year period from January 1999 to November 2019, with 27% of included trials reporting seven

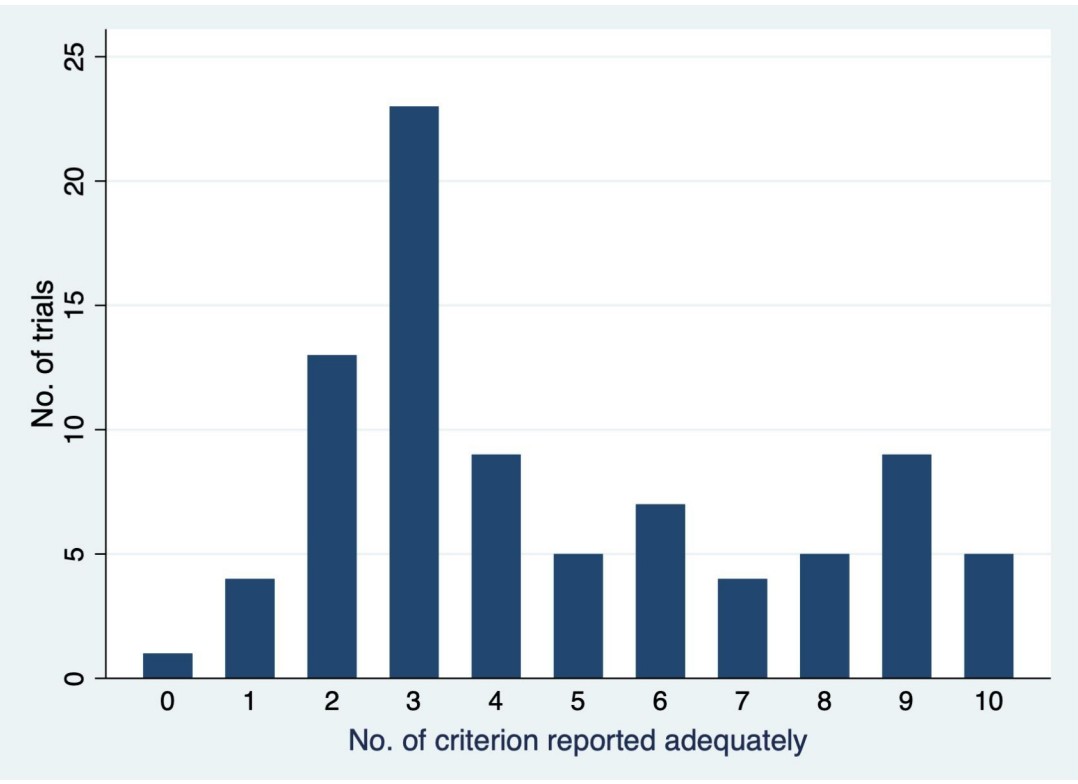

**Fig 2. Total number of criterion which were reported adequately in all trials.**

**Table 4. Multivariable logistic regression.**

| Factors associated with adequate quality reporting | | No. of trials with adequate quality reporting (%) | No. of trials with inadequate quality reporting (%) | Odds ratio | 95% CI | P value |
|---|---|---|---|---|---|---|
| Cooperative Group | No | 3 (9%) | 32 (91%) | Reference | | |
| | Yes | 20 (40%) | 30 (60%) | 4.65 | 1.13 to 19.11 | 0.033 |
| QA results published | No | 13 (18%) | 58 (82%) | Reference | | |
| | Yes | 10 (71%) | 4 (29%) | 8.50 | 1.87 to 38.56 | 0.006 |
| Risk of Bias in methodological quality | Low risk | 22 (36%) | 39 (64%) | Reference | | |
| | Some concerns | 1 (4%) | 23 (96%) | 0.09 | 0.01 to 0.81 | 0.031 |

or more quality measure criteria adequately. The total radiation dose and fractionation schedule criteria were reported adequately in majority of the trials. The organ at risk dose constraints and deviation in radiation treatment planning and delivery criteria were reported adequately in less than one third of the included trials.

The results of this study are consistent with other published studies. Bekelman et al assessed 61 radiotherapy trials of lymphoma for adequate reporting of six RT criteria namely target volume, radiation dose, fractionation, radiation prescription, quality assurance process use and adherence to quality assurance (i.e. reporting of major or minor deviations) [4]. They found that the reporting of these six RT criteria to be deficient. Less than a third of the trials reported the target volume, radiation prescription, quality assurance process use and adherence to quality assurance process adequately. Tseng et al assessed 67 radiotherapy trials of head and neck cancers for adequate reporting of the same six RT criteria and similarly found that less than a third of the trials reported target volume, radiation prescription and adherence to quality assurance adequately [5]. Soon et al evaluated 59 radiotherapy trials of prostate cancer for adequate reporting. It was reported that only one-third of the trials reported organ at risk dose constraints, simulation procedures and adherence to quality assurance process adequately [6]. In another study looking at the quality of reporting in radiotherapy trials of lung cancer, it was found that 27% of the trials reported seven criteria or more adequately [7].

There are two possible explanations for the inadequate reporting of CR technique in our study. One is the lack of consensus guidelines to inform the investigators on the minimum standard of reporting of RT technique in clinical trials. Although the CONSORT statement provide guidance for reporting of non-pharmacological treatment in randomized trials, there is a lack of specific recommendations in the CONSORT statement for reporting of radiotherapy treatments [9]. Secondly, there is probably a lack of awareness amongst research groups, on the recommendations for reporting of radiotherapy techniques in clinical trials. Bentzen has recommended a set of minimum criteria to follow for reporting of radiotherapy trials and Nilsson et al has produced a template on writing radiotherapy protocols for clinical trials [10, 11]. Unfortunately, their recommendations have not gained worldwide acceptance or being incorporated into the CONSORT statement.

It is not surprising to observe that cooperative group trials, trials that published its trial protocols and QA results are more likely to have more relevant details on CR technique. For example, NRG Oncology has established a center for innovation in radiation oncology to foster collaboration between cooperative groups and to standardize the description of RT techniques in the clinical trial protocols of various cooperative groups [12]. Trials that published its trial protocols and QA results are more likely to provide more details to the readers on RT technique and RTQA processes.

This study has a few methodological strengths. Firstly, we employed published tools to evaluate the quality of CR reporting [6]. This will help facilitate the comparison of our study results with the previous studies. Secondly, only randomized trials were selected for this study because we believe that randomized trials has the most important role in influencing clinical practice compared to other study designs, and the practicing radiation oncologists are more likely to adopt the CR treatment technique described in these trials.

This study has a few methodological limitations. Firstly, the sample size is fairly small, hence we are unable to assess the effect of more than three variables in our multivariate model. However, our results are consistent with previous studies, thus providing support to this study's findings. Secondly, not all trials referenced their trial reports or included their trial protocols as supplementary materials for assessment, hence the quality of CR reporting may possibly be more adequate if all the trials protocols are available. This was observed in the multivariable logistic regression analysis which showed that trials with published trial protocols were more likely to have higher quality in the reporting of CR compared to trials without a published trial protocol. We recognized that journals have set a maximum word limit for the manuscript and it may be challenging for the investigators to provide detailed descriptions of these RT quality measures in the primary trial report. Nevertheless, we encourage the investigators to at least provide a short description of the RT treatment technique in the primary trial report and with more detailed description in the supplementary materials. Thirdly, we searched only two databases mainly MEDLINE via PubMed and EMBASE for eligible studies. It is possible that our search is not sufficiently comprehensive and some eligible studies may have been missed.

These study findings suggest that there is a strong need to have a consensus for the reporting of radiation therapy technique in randomized trials. The complexity of radiotherapy treatment has increased over the years. Computed tomography (CT) planning is the critical aspect for delivery of radiation to the primary brain tumors. CT planning allows us to identify the gross tumor volumes as well as the organs at risk accurately. Besides assessing coverage of the gross tumor, we also need to assess and report the organs at risk constraints (such as brain stem, optic chiasm, optic nerves) using dose volume histograms. Given the increased complexity of CR treatment delivery, it is even more important for the new trials to provide enough information on the CR techniques.

## Conclusions

In conclusion, the quality of reporting of CR technique in the majority of randomized trials of primary brain tumors is mostly inadequate. Formal consensus reporting guidelines for radiotherapy treatment in trials from the CONSORT group are needed to improve the quality of CR technique reporting.

## Supporting information

**S1 Table. Search strategy.**
(DOCX)

**S2 Table. List of included studies.**
(DOCX)

**S3 Table. Characteristics of included studies.**
(DOCX)

**S4 Table. Risk of bias in the methodological quality.**
(DOCX)

**S5 Table. Adequate reporting of each criterion.**
(DOCX)

**S1 Data.**
(XLSX)

## Author Contributions

**Conceptualization:** Teng Hwee Tan, Yu Yang Soon, Jeremy Chee Seong Tey.

**Data curation:** Teng Hwee Tan, Desiree Chen, Yu Yang Soon, Jeremy Chee Seong Tey.

**Formal analysis:** Desiree Chen.

**Methodology:** Teng Hwee Tan, Yu Yang Soon, Jeremy Chee Seong Tey.

**Project administration:** Yu Yang Soon.

**Writing – original draft:** Teng Hwee Tan, Desiree Chen, Yu Yang Soon, Jeremy Chee Seong Tey.

**Writing – review & editing:** Teng Hwee Tan, Desiree Chen, Yu Yang Soon, Jeremy Chee Seong Tey.

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
