## [Decision Letter · Decision Letter 0]

4 Jun 2020

PONE-D-20-01058

Quality of reporting of cranial irradiation techniques in randomized controlled trials of primary brain tumors:  a systematic review

PLOS ONE

Dear Dr. Soon,

Thank you for submitting your manuscript to PLOS ONE. After careful consideration, we feel that it has merit but does not fully meet PLOS ONE’s publication criteria as it currently stands. Therefore, we invite you to submit a revised version of the manuscript that addresses the points raised during the review process.

Specifically, the reviewers have raised overlapping concerns about the statistical methodology and reporting of the search strategy in the manuscript.

We look forward to receiving your revised manuscript.

Kind regards,

Richard Hodge

Associate Editor

PLOS ONE

Journal Requirements:

2. Upon revision, please ensure that inclusion and exclusion criteria for the selected studies are clearly stated in the Methods section. Please also give a rationale for using 7 criteria as a cut-off for good reporting of CR technique. Please also list which criteria were met for each study considered.

3. We noticed you have some minor occurrence(s) of overlapping text with the following previous publication(s), which needs to be addressed:

https://doi.org/10.1016/j.ijrobp.2017.06.862

https://doi.org/10.1097/MD.0000000000016124

https://doi.org/10.1200/JCO.2008.19.4522

In your revision ensure you cite all your sources (including your own works), and quote or rephrase any duplicated text outside the Methods section. Further consideration is dependent on these concerns being addressed.

4. Please organise your Abstract in subheadings.

Reviewers' comments:

Reviewer's Responses to Questions

**Comments to the Author**

1. Is the manuscript technically sound, and do the data support the conclusions?

Reviewer #1: Partly

Reviewer #2: Yes

Reviewer #3: Yes

2. Has the statistical analysis been performed appropriately and rigorously? 

Reviewer #1: No

Reviewer #2: No

Reviewer #3: Yes

3. Have the authors made all data underlying the findings in their manuscript fully available?

Reviewer #1: No

Reviewer #2: No

Reviewer #3: Yes

4. Is the manuscript presented in an intelligible fashion and written in standard English?

Reviewer #1: No

Reviewer #2: Yes

Reviewer #3: Yes

5. Review Comments to the Author

Reviewer #1: This meta-analysis is about "Quality of reporting of cranial irradiation techniques in randomized controlled trials of primary brain tumors: a systematic review".

In my opinion, the article does not have the necessary quality to be accepted

Reviewer #2: Tan and colleagues aimed to evaluate the quality of reporting of cranial irradiation techniques in randomized controlled trials in primary brain tumor patients. They found that the quality of reporting is suboptimal in many studies, warranting the introduction of guidelines on reporting.

1. Introduction: What is the rationale for determining predictors for the quality of reporting? How will this be used? This should be described in the introduction.

2. Method: Were only studies with malignant primary brain tumors included? Or also benign tumors? Also, it should be described if the focus is on pediatric or adult patients, or both? In short, the specific in- and exclusion criteria should be reported.

3. Methods: What was the reason to exclude PCNSL? I understand that brain metastases are excluded, as they are not primary brain tumors, but it is unclear why studies including PCNSL patients are excluded.

4. Methods: Why were only MeSH terms included in the search strategy? Also, only two databases were searched. This should be recognized as a limitation.

5. Methods: The authors state that the 10 quality criteria have been described in previous publications. What are the references? And were these 10 criteria also used in the previous reviews (as described in the introduction) to assess the quality of reporting of irradiation techniques? What were similarities and differences? And if not equal, why were different methods used?

6. Methods: Was each variable first tested in a univariable model before entering in the multivariable model? If not, this should be explained.

7. Methods: What is the rationale for including ‘bias in reporting the primary efficacy endpoint’ or ‘bias in reporting of primary toxicity endpoint’ as predictors for good quality of reporting of RT techniques? And not characteristics such as tumor type?

8. Methods: only 23 studies were classified as ‘adequate quality in reporting’. This means that only 2 (and max 3) variables can be included in the multivariable model, otherwise it would be overfitted. In this study, more than 3 variables seem to be included, limiting the reliability of the model.

9. Results: Data on each eligible study was collected, including characteristics such as patient population. It would be helpful to add characteristics (tumor type, number of patients, intervention, sociodemographics) to the supplemental table.

10. Results: Bias in reporting the primary efficacy endpoint was predictive of good quality reporting of RT techniques? This is counterintuitive: if they would not properly report the primary endpoints, why would they properly report information on RT techniques? I am not convinced by the explanation provided by the authors.

11. Results: The odds ratios 6.67 and 5.17 are converted to percentages. However, shouldn’t these be 567% and 417% instead of 667% and 517% (and OR of 2 is a 100% increase in chance, not 200%)?

12. Discussion: The authors argue in the introduction that information on the radiation techniques should be reported adequately in RCTs, to reduce the risk of deviation in daily clinical practice. In this review, all type of primary brain tumors are combined. I wonder, however, if there are differences in trials for different types of brain tumors? And what about trials that were practice-changing versus those who were not? It would be informative if these subanalysis would be performed.

13. Discussion: as a limitation, a small sample size is mentioned, and therefore no definitive conclusion can be drawn. A definitive conclusion on what?

Reviewer #3: METHODS:

Trial eligibility criteria:

1. Specify study characteristics (e.g., PICOS, length of follow-up) and report characteristics (e.g., years considered, language, publication status) used as criteria for eligibility, giving rationale.

2. It should be need to present full electronic search strategy for at least one database, including any limits used, such that it could be repeated.

3. Why the authors didn't use other major database such as ISI, Scopus?

Selection of trials and Data extraction:

4. The full text article of any trial that appeared to meet the inclusion criteria was retrieved for closer inspection. Disagreements were resolved by consensus. If the agreement rate was assessed? Was the KAPPA statistics?

5. Describe methods used for assessing risk of bias of individual studies (including specification of whether this was done at the study or outcome level), and how this information is to be used in any data synthesis

Results:

5. Describe the result of assessing risk of bias of individual studies?

6. Table 4: p-values must be reported as three digit.

6. PLOS authors have the option to publish the peer review history of their article (what does this mean?). If published, this will include your full peer review and any attached files.

Reviewer #1: No

Reviewer #2: No

Reviewer #3: No

---

## [Author Response · Author response to Decision Letter 0]

24 Jul 2020

Please see file labeled as 'Response to reviewers'

---

## [Decision Letter · Decision Letter 1]

3 Sep 2020

PONE-D-20-01058R1

Quality of reporting of cranial irradiation techniques in randomized controlled trials of primary brain tumors:  a systematic review

PLOS ONE

Dear Dr. Soon:

Thank you for submitting your manuscript to PLOS ONE. After careful consideration, we feel that it has merit but does not fully meet PLOS ONE’s publication criteria as it currently stands. Therefore, we invite you to submit a revised version of the manuscript that addresses the points raised during the review process.

There have been three reviews:  one to accept, one for minor revisions, one to reject.  Please attempt to address all of these in the next submission.  I will note that the "reject" had few comments. 

We look forward to receiving your revised manuscript.

Kind regards,

Gayle E. Woloschak, PhD

Academic Editor

PLOS ONE

Additional Editor Comments (if provided):

There have been three reviews: one to accept, one for minor revisions, one to reject. Please attempt to address all of these in the next submission. I will note that the "reject" had few comments.

Reviewers' comments:

Reviewer's Responses to Questions

**Comments to the Author**

1. If the authors have adequately addressed your comments raised in a previous round of review and you feel that this manuscript is now acceptable for publication, you may indicate that here to bypass the “Comments to the Author” section, enter your conflict of interest statement in the “Confidential to Editor” section, and submit your "Accept" recommendation.

Reviewer #1: (No Response)

Reviewer #2: All comments have been addressed

Reviewer #3: (No Response)

2. Is the manuscript technically sound, and do the data support the conclusions?

Reviewer #1: Partly

Reviewer #2: Yes

Reviewer #3: Yes

3. Has the statistical analysis been performed appropriately and rigorously? 

Reviewer #1: No

Reviewer #2: Yes

Reviewer #3: Yes

4. Have the authors made all data underlying the findings in their manuscript fully available?

Reviewer #1: No

Reviewer #2: Yes

Reviewer #3: Yes

5. Is the manuscript presented in an intelligible fashion and written in standard English?

Reviewer #1: No

Reviewer #2: No

Reviewer #3: Yes

6. Review Comments to the Author

Reviewer #1: Thanks for submit your revised version of manuscript about "Quality of reporting of cranial irradiation techniques in randomized controlled trials of primary brain tumors: a systematic review"

In my opinion you did not write your article correctly and it should be rewrite principally.

Reviewer #2: The authors have addressed all my comments, which helped to improve the manuscript. The methodology is more clearly described, which facilitates interpretation of the results.

My only concern is the English language. I would advise to involve a native speaker for grammatical edits.

Reviewer #3: (No Response)

7. PLOS authors have the option to publish the peer review history of their article (what does this mean?). If published, this will include your full peer review and any attached files.

Reviewer #1: No

Reviewer #2: No

Reviewer #3: No

---

## [Author Response · Author response to Decision Letter 1]

8 Oct 2020

Please attached file named as response to reviewers

---

## [Decision Letter · Decision Letter 2]

19 Oct 2020

Quality of reporting of cranial irradiation techniques in randomized controlled trials of primary brain tumors:  a systematic review

PONE-D-20-01058R2

Dear Dr. Soon:

We’re pleased to inform you that your manuscript has been judged scientifically suitable for publication and will be formally accepted for publication once it meets all outstanding technical requirements.

Kind regards,

Gayle E. Woloschak, PhD

Section Editor

PLOS ONE

Additional Editor Comments (optional):

Thank you for addressing the concerns of the reviewers.

Reviewers' comments:

Reviewer's Responses to Questions

**Comments to the Author**

1. If the authors have adequately addressed your comments raised in a previous round of review and you feel that this manuscript is now acceptable for publication, you may indicate that here to bypass the “Comments to the Author” section, enter your conflict of interest statement in the “Confidential to Editor” section, and submit your "Accept" recommendation.

Reviewer #2: All comments have been addressed

Reviewer #3: All comments have been addressed

2. Is the manuscript technically sound, and do the data support the conclusions?

Reviewer #2: Yes

Reviewer #3: Yes

3. Has the statistical analysis been performed appropriately and rigorously? 

Reviewer #2: Yes

Reviewer #3: Yes

4. Have the authors made all data underlying the findings in their manuscript fully available?

Reviewer #2: Yes

Reviewer #3: Yes

5. Is the manuscript presented in an intelligible fashion and written in standard English?

Reviewer #2: Yes

Reviewer #3: Yes

6. Review Comments to the Author

Reviewer #2: The authors have rewritten several parts of the manuscript, and now reflects adequate English language.

Reviewer #3: Dear editor,

The revised manuscript (entitled: Quality of reporting of cranial irradiation techniques in randomized controlled trials of primary brain tumors: a systematic review) was rechecked, all comments have been addressed.

7. PLOS authors have the option to publish the peer review history of their article (what does this mean?). If published, this will include your full peer review and any attached files.

Reviewer #2: No

Reviewer #3: No

---

## [Editor Report · Acceptance letter]

21 Oct 2020

PONE-D-20-01058R2 

Quality of reporting of cranial irradiation techniques in randomized controlled trials of primary brain tumors: a systematic review 

Dear Dr. Soon:

I'm pleased to inform you that your manuscript has been deemed suitable for publication in PLOS ONE. Congratulations! Your manuscript is now with our production department. 

Kind regards, 

on behalf of

Dr. Gayle E. Woloschak 

Section Editor

PLOS ONE